

# Lagrangian detection of precipitation moisture sources for an arid region in northeast Greenland: relations to the North Atlantic Oscillation, sea ice cover and temporal trends from 1979 to 2017

Lilian Schuster[1], Fabien Maussion[1], Lukas Langhamer[2], and Gina E. Moseley[3]

[1]Department of Atmospheric and Cryospheric Sciences (ACINN), University of Innsbruck, Innsbruck, Austria
[2]Department of Geography, Faculty of Mathematics and Natural Science, Humboldt University of Berlin, Berlin, Germany
[3]Institute of Geology, University of Innsbruck, Innsbruck, Austria

**Correspondence:** Lilian Schuster (lilian.schuster@student.uibk.ac.at)

**Abstract.** Temperature in northeast Greenland is expected to rise at a faster rate than the global average as consequence of anthropogenic climate change. Associated with this temperature rise, precipitation is also expected to increase as a result of increased evaporation from a warmer and ice-free Arctic Ocean. In recent years, numerous palaeoclimate projects have begun working in the region with the aim of improving our understanding of how this highly-sensitive region responds to a warmer

5 world. However, a lack of meteorological stations within the area makes it difficult to place the palaeoclimate records in the context of present-day climate. This study aims to improve our understanding of precipitation and moisture source dynamics over a small arid region located at $80°$ N in northeast Greenland. The origin of water vapour for precipitation over the study region is detected by a Lagrangian moisture source diagnostic, which is applied to reanalysis data from the European Centre for Medium-Range Weather Forecasts (ERA-Interim) from 1979 to 2017. While precipitation amounts are relatively constant

10 during the year, the regional moisture sources display a strong seasonality. The most dominant winter moisture sources are the North Atlantic above $45°$ N and the ice-free Atlantic sector of the Arctic Ocean, while in summer the patterns shift towards local and north Eurasian continental sources. During the positive phases of the North Atlantic Oscillation (NAO), evaporation and moisture transport from the Norwegian Sea is stronger, resulting in larger and more variable precipitation amounts. Testing the hypothesis that retreating sea ice will lead to increase in moisture supply remains challenging based on our data. However, we

15 found that moisture sources are increasing in case of retreating sea ice for some regions, in particular in October to December. Although the annual mean surface temperature in the study region has increased by $0.7\,°\mathrm{C}\,\mathrm{dec}^{-1}$ (95% confidence interval $[0.4, 1.0]\,°\mathrm{C}\,\mathrm{dec}^{-1}$) according to ERA-Interim data, we do not detect any change in the amount of precipitation with the exception of autumn where precipitation increases by $8.2\,[0.8, 15.5]\,\mathrm{mm}\,\mathrm{dec}^{-1}$ over the period. This increase is consistent with future predicted Arctic precipitation change. Moisture source trends for other months and regions were non-existent or small.



## 1 Introduction

The Arctic region is known from both observational and modelling studies to be highly-sensitive to changes in climate. This high sensitivity is the result of Arctic amplification; a process in which positive feedbacks act to amplify changes as compared to the rest of the Northern Hemisphere. Observational studies have shown the effect of Arctic amplification during both former warm climates such as Quaternary interglacials, as well as cold climates such as Quaternary glacials (e.g., Dahl-Jensen et al., 1998; Miller et al., 2010). In the future, under a regime of increasing atmospheric greenhouse gas concentrations, surface air temperature rise in the Arctic is also expected to be amplified (e.g., Serreze and Barry, 2011) predominantly as a result of surface-albedo changes (e.g., Serreze et al., 2009), oceanic heat loss (e.g., Screen and Simmonds, 2010) and infrared-radiation feedbacks (e.g., Bintanja and Van der Linden, 2013). Many lines of evidence already suggest that Arctic amplification is a feature of Earth's changing climate (e.g., Serreze et al., 2009). Between 1875 to 2008, surface air temperature north of $60°\,N$ increased at twice the pace of the Northern Hemisphere average (e.g., Bekryaev et al., 2010), with the winter season being the most affected because of the delayed onset of sea ice resulting in a loss of heat from the open ocean to the atmosphere (e.g., Screen and Simmonds, 2010; Bintanja and Van der Linden, 2013). Recent warming within the Arctic has not, however, been homogeneous, and in some parts average winter surface temperatures have risen by as much as $4\,°C$ to $5\,°C$ over the last 50 years (GISTEMP Team, 2016; Shepherd, 2016). The areas that were most affected included northwest North America and northeast Greenland.

Amplified precipitation changes are expected to accompany amplified temperature changes (e.g., Collins et al., 2013; Bintanja and Selten, 2014; Bintanja and Andry, 2017) and indeed evidence already exists to suggest an increase in Arctic precipitation during the last century (e.g., Kattsov and Walsh, 2000). Unfortunately, a lack of stations north of $70°\,N$ does, however, limit our understanding of the evolving hydrological regime (e.g., Kattsov and Walsh, 2000; Kurita, 2011; Bintanja and Selten, 2014). The majority of models agree that in the future, mean annual precipitation will increase in the mid-latitudes and polar regions, driven largely by the increase in surface temperature (e.g., Collins et al., 2013). Rain is predicted to become the dominant form of precipitation in the Arctic in all areas apart from the interior of the Greenland Ice Sheet (e.g., Bintanja and Andry, 2017), and by the end of this century, the greatest changes in Arctic precipitation, which could be as much as $50\,\%$ in an RCP 8.5 scenario, are found over the Arctic Ocean and northeast Greenland (Bintanja and Selten, 2014), with the majority of change occurring in the late autumn–winter months. Models predict that moisture transport towards the Arctic will increase in the future reaching a maximum during summer months when meridional temperature and moisture gradients are at their maximum (e.g., Bintanja and Selten, 2014). However, whilst the absolute values of moisture transported to the Arctic are expected to increase, the relative contribution of this source will diminish in comparison to locally sourced moisture, which will be enhanced due to increased surface evaporation from open ice-free Arctic waters in late autumn–winter (e.g., Bintanja and Selten, 2014).

Models for the coming decades indicate that within the Arctic, northeast Greenland is one of the most sensitive terrestrial areas to changing temperatures and precipitation (Koenigk et al., 2013; Bintanja and Selten, 2014; GISTEMP Team, 2016; Shepherd, 2016). Furthermore, climate and palaeoclimate research activities in northeast Greenland have increased in recent



years in response to various needs to improve fundamental understanding of the climate and environment of this region.

For example, several projects (e.g., NEGIS project, 2020; EastGRIP, 2018) are researching the dynamics of the Northeast Greenland Ice Stream (NEGIS), which is an important component of the Greenland Ice Sheet that delivers ice into a part of the Atlantic Ocean that is sensitive to freshwater forcing (NEGIS project, 2020). Of these projects, "EastGRIP" has been drilling and analysing an ice core in order to improve understanding of ice stream dynamics and their role in future sea-level change (EastGRIP, 2018), whilst the "NEGIS Project" has been researching ocean sediment cores in order to better understand the

response of the NEGIS to increased temperatures. Elsewhere in northeast Greenland, modelling research has shown that the region is expected to undergo the greatest expansion of supraglacial lakes during the 21$^{st}$ century as compared to the rest of the ice sheet (Ignéczi et al., 2016), whilst the role that surface meltwater plays in recharging subglacial lakes in northeast Greenland has also been shown to be important (Willis et al., 2015). Simulations of the Greenland Ice Sheet during the last interglacial suggest that the northeast sector is most vulnerable to increases in temperature because many years of accumulation are lost

creating a strong ice-elevation feedback that is further hampered by low accumulation rates (Born and Nisancioglu, 2012). Finally, new research into speleothems in northeast Greenland by the Greenland Caves Project (2015) is aiming to improve knowledge of past climates and environments in this region in a warmer world (Moseley, 2016).

In summary, northeast Greenland is known to be highly sensitive to global climate changes, which has resulted in an increase in fundamental research in recent years. Despite this, a distinct knowledge gap exists with regards to the present-day

climatology of northeast Greenland. Climatological moisture source studies have thus far tended to concentrate on the Greenland Ice Sheet (30 selected winter months, Sodemann et al., 2008a, b), whereas Nusbaumer et al. (2019) separated Greenland moisture sources into four sectors but focused mainly on northwest Greenland. Here we aim to address this knowledge gap through investigation of the climatology of precipitation and its moisture sources over a 39-year period (1979 to 2017) in the presently arid region located at the study site of the Greenland Caves Project (80° N, 22° W, 740 m a.s.l) in northeast Greenland

(Fig. 1, 2). The results will, however, have wider implications for other studies working in the area. Moisture sources are diagnosed from the European Centre for Medium-Range Weather Forecasts (ECMWF) ERA-Interim reanalysis dataset applying the Lagrangian moisture source diagnostic by Sodemann et al. (2008a) with the adjustment of the planetary boundary layer (PBL) height according to Langhamer et al. (2018). In particular, we aim to investigate: (1) the annual cycle of precipitation in the region and its moisture sources; (2) the inter-annual variability of moisture sources and their relation with the North

Atlantic Oscillation (NAO) and sea ice cover variability, and; (3) whether the precipitation characteristics and moisture sources have shown any significant changes in recent decades.

## 2 Data and Methods

### 2.1 Reanalysis data

In this study, reanalysis data from the European Centre for Medium-Range Weather Forecasts (ERA-Interim) was used (Berris-

ford et al., 2011; Dee et al., 2011; Owens and Hewson, 2018) for both precipitation and moisture source estimates. This product was chosen because a consistent model without spatial or temporal gaps was needed to apply the moisture source diagnostic





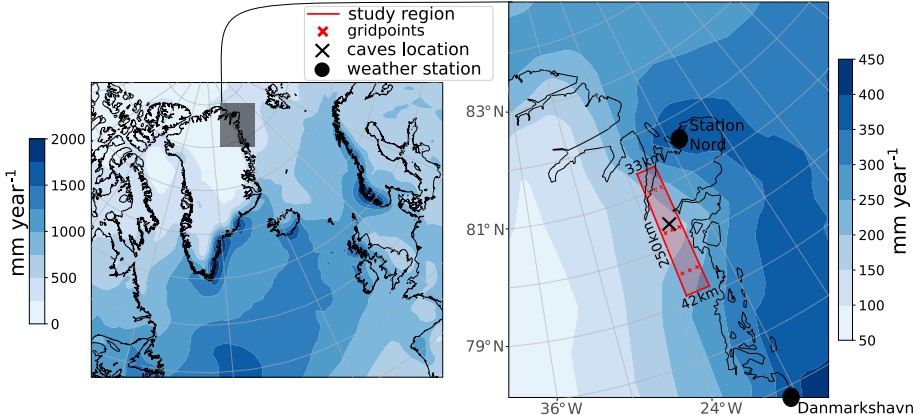

**Figure 1.** Average of yearly ERA-Interim precipitation (1979–2018). The study region is depicted with the nine gridpoints located between 22.5° W and 21° W and between 79.5° N and 81° N. The exact location of the studied caves is (21.7419° W, 80.3745° N). Average precipitation in the study region is 207 mm year$^{-1}$ (95 % confidence interval of [192, 224] mm year$^{-1}$).

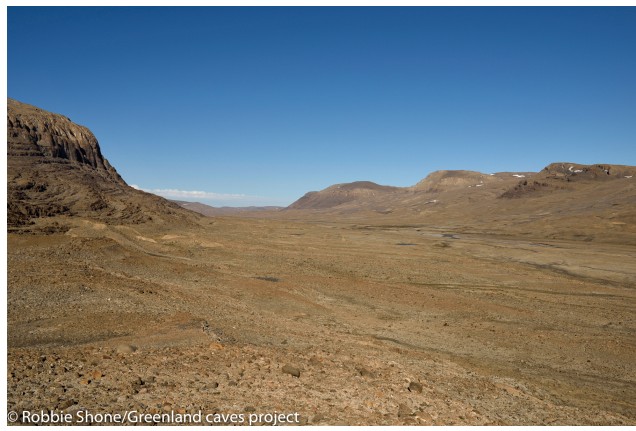

**Figure 2.** Photograph from the arid study region near to the location of the caves, ©Robbie Shone/Greenland Caves Project (August 2015).

(see Sect. 2.2). ERA-Interim has a fully revised humidity scheme and higher spatial resolution (∼79 km) than ERA-40, that was used by Sodemann et al. (2008a) to compute Greenland winter precipitation sources. The even newer ERA5 reanalysis data was not yet available at the time we conducted these analyses. The study region (21° W–22.5° W, 79.5° N–81° N, Fig. 1)

consists of nine gridpoints with a horizontal resolution of 0.75°, which are located around the caves in northeast Greenland. Several gridpoints were chosen to smooth out local inhomogeneities. The ERA-Interim dataset is used in the time span of February 1979–May 2017 for which we computed the Lagrangian diagnostics. For the temporal precipitation trends and for the total average precipitation the time period was extended to January 1979–December 2018. For the estimates of moisture source trends, the period was shortened to January 1980–December 2016 in order to cover full years only. To estimate the

moisture sources of the study region by the Lagrangian moisture source diagnostic (Sect. 2.2), 6-hourly specific humidity,





3D-wind field, surface pressure, PBL height and two metre temperature were used. Moisture sources over land and ocean are distinguished by using the *land/sea mask* of ERA-Interim on the same $0.75°$ grid. For each gridpoint, the monthly sea ice area was computed by multiplying the ERA-Interim *sea ice fraction* (0–1) by the latitudinally-weighted gridpoint area. To classify gridpoints into land, ocean and sea ice, a threshold of 0.5 was set for the fractional *land/sea mask* and *sea ice fraction*.

In addition, relations between precipitation and its moisture sources to other ERA-Interim parameters were examined. These are the sea ice area, the mean $500\,\mathrm{hPa}$ geopotential height, and the vertically integrated water vapour transport (sum of the integrated northward and eastward cloud liquid, cloud frozen, and water vapour transport). To relate precipitation and moisture source variability to large-scale teleconnection patterns, we computed correlations to the monthly NAO index data from the National Oceanic and Atmospheric Administration climate prediction centre (NOAA, 2020).

## 105  2.2  Trajectory calculation and Lagrangian moisture source diagnostic

To compute the motion of air parcels, 15-day backward trajectory calculations by the Lagrangian Analysis tool LAGRANTO version 2.0 (Sprenger and Wernli, 2015), first version by Wernli and Davies (1997), were realised for every six hours from February 1979 to May 2017. Trajectories start at the node of the $0.75°$ regular grid of the study region (9 gridpoints, Fig. 1) on 11 vertical levels from the surface to a height of $500\,\mathrm{hPa}$ ($\Delta p = 49.9\,\mathrm{hPa}$). This corresponds to 99 trajectories per time step.

In the next step, the trajectories that aren't leading to precipitation in the study region were filtered out. The requirements for the selected trajectories were that relative humidity exceeded $80\,\%$ and specific humidity ($q$) decreased in the last time step (Sodemann et al., 2008a).

Evaporation and precipitation of precipitation-trajectories are identified by temporal changes in specific humidity ($\Delta q$). Using the assumption of a well-mixed PBL, the moisture content of air parcels increases within the PBL in case of a positive

$\Delta q$. Moisture uptakes that occur above the PBL, however, are detached from the surface and are explained by physical or numerical processes, e.g., convection, evaporation of precipitating hydro-meteors, change of liquid water content, or ice water content, subgrid-scale turbulent fluxes, numerical diffusion, and errors, or physical inconsistencies (Sodemann et al., 2008a). Along each trajectory, moisture uptake locations inside the PBL are weighted by their contribution to the total precipitation in the study region by taking en route precipitation into account. Each moisture uptake is interpolated on a $1°$ grid and we

calculate the monthly means on this basis.

This Lagrangian approach of Sodemann et al. (2008a) is suitable for our study as it gives similar moisture source regions and needs lower computation cost than a more complex Eulerian approach (Winschall et al., 2014, for a case study in Europe). The marine PBL height is often underestimated in numerical weather prediction models (Zeng et al., 2004). Therefore, the threshold for a moisture source location inside the PBL height is lifted by a factor of 1.5. Similar to Langhamer et al. (2018),

the height of the PBL was converted in our study into pressure coordinates by applying the barometric formula with surface pressure and temperature as free variables (and a constant temperature lapse rate of $0.0065\,\mathrm{K\,m^{-1}}$).

A measure for the performance of the method is shown in Fig. 3. The Lagrangian evaporation sum that contributed to precipitation in the study region correlates very well with the total precipitation over the study region from ERA-Interim, indicating that the method is able to reproduce the variability of precipitation. $48\,\%$ of the total moisture sources could be





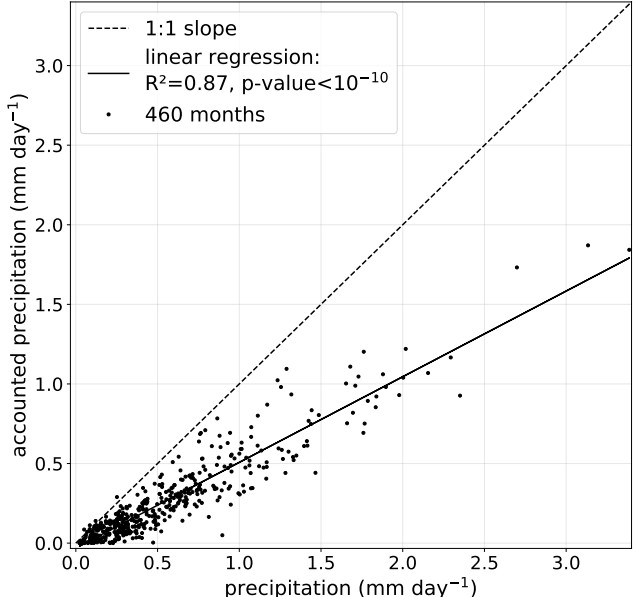

**Figure 3.** Scatterplot of accounted precipitation (summed-up attributed contributing evaporation from the Lagrangian moisture source diagnostic, mean: $0.27\,[0.25, 0.30]\,\mathrm{mm\,day^{-1}}$) against total precipitation in study region (mean: $0.56\,[0.52, 0.61]\,\mathrm{mm\,day^{-1}}$) for all months (February 1979–May 2017 using ERA-Interim).

assigned to specific evaporation locations with the applied Lagrangian moisture source diagnostic. For comparison, similar studies by Sodemann and Zubler (2010) in the European Alps and Langhamer et al. (2018) in Patagonia reached 50% and 71% attribution, respectively. The remaining moisture sources could not be identified to evaporation at the surface (moisture uptake above PBL) or were unidentifiable. There is no clear annual cycle visible in the attribution: the fraction ranges from a minimum of $41\,\%$ in August to a maximum of $57\,\%$ in June (Fig. 4). Specifically in summer, precipitation in the study region

varies more than its attributed moisture sources. We discuss the possible implications of these uncertainties in Sect. 5.4.

### 2.3 Statistical methods

To compute confidence intervals of our trends and averages, we estimate the $95\,\%$ confidence intervals of the mean or median (significance level of 0.05) without assuming a parametric distribution by using the bootstrapping method (Wilks, 2011). This is done because some subsets of daily precipitation averaged over a month as well as other related parameters reject the null

hypothesis that their distributions are drawn from a normal distribution using a Shapiro-Wilk normality test (Wilks, 2011). To describe the uncertainties, these $95\,\%$ confidence intervals of the values are indicated in brackets [ , ] behind the actual value. Therefore, significant differences in e.g., the mean of two values occur at the $5\,\%$ significance level if the $95\,\%$ confidence intervals do not overlap.

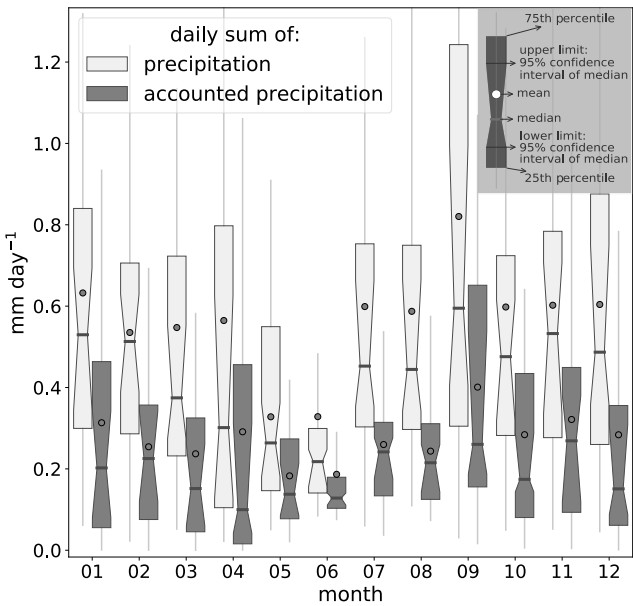

**Figure 4.** Boxplots show annual cycle of precipitation and accounted precipitation (summed-up attributed contributing evaporation from the Lagrangian moisture source diagnostic) in the study region for all months (February 1979–May 2017 using ERA-Interim).

For the climate indices, months with exceptionally low NAO values (below the 25 % percentile) are herein referred to as NAO−. Months with exceptionally high NAO values (above the 75 % percentile) are herein referred to as NAO+. NAO values that fall between the lower and upper quartile are referred as NAO neutral. This classification is either done for all months together, or in the case of the annual cycle separate thresholds for each month of the year were computed. To measure the association between two variables, we mostly use the Spearman's rank correlation coefficient instead of the Pearson correlation coefficient, as it reflects the strength of a monotonic relationship instead of a linear relationship, and is therefore more robust to outliers (Wilks, 2011).

## 3 Precipitation and moisture source characteristics

### 3.1 Mean and Annual cycle of precipitation

According to ERA-Interim, for the period February 1979–May 2017, the mean precipitation is 207 [192, 224] mm year$^{-1}$ averaged over the study region. At the nearest gridpoint to the caves, it is slightly drier with 171 [158, 185] mm year$^{-1}$. The North Atlantic cyclone track decays northward (Serreze and Barry, 2014), and up to ten times fewer precipitation occurs in northeast Greenland than on the southeast coast (Fig. 1). As is typical for regions with little precipitation (e.g., Pendergrass and Knutti, 2018), less events bring most of the total precipitation. On average, the five wettest days in a year produce 24 % and around 16 days produce 50 % of the total annual precipitation in the study region. Precipitation can happen throughout the year, but





May and June are slightly drier on average whereas September is wettest (Fig. 4). September is the wettest month for 9 of 40
years, June is the driest month for 6 of 40 years, and April is the driest month for 8 of 40 years. September (as the wettest
month) has the greatest variability (interquartile range of 0.30–1.24 mm day$^{-1}$), whereas June (as the driest month) displays
the least variability (interquartile range of 0.14–0.30 mm day$^{-1}$). April also shows a large variability (interquartile range of
0.10–0.80 mm day$^{-1}$) and is the month with the most positively skewed monthly precipitation distribution. Generally, precip-
itation over the Atlantic Arctic sector is stronger in winter months due to the enhanced North Atlantic cyclone track over the
relatively warm open water and the moisture flux convergence specifically near to the Icelandic Low. For continental areas
above 60° N, however, most precipitation occurs in July, August and September (ERA-Interim and ASRv1, Bromwich et al.,
2016). This can be explained by higher cyclone and frontal activity in summer because of heating contrasts between snow-free
land and snow areas (Serreze and Barry, 2014).

### 3.2 Mean and Annual cycle of moisture sources

While precipitation amounts are relatively constant during the year, the corresponding contributing moisture sources display
a strong seasonality in magnitude and location (Fig. 5). In winter, most moisture sources are located over the North Atlantic
above 45° N and the ice-free Atlantic sector of the Arctic Ocean with a maximum between Scandinavia and Svalbard. This
maximum is most pronounced in January and then gradually diminishes until May. Starting with May, local moisture sources
begin to contribute to precipitation and peak in June. In July, moisture sources seem to come mostly from land areas over the
north Eurasian continent. September has the minimum amount of sea and land ice and represents a transitional phase, where
there seems to be both land sources over Scandinavia and the majority of ocean sources over the North Atlantic. This could
be a possible indicator why precipitation is strongest in September (Fig. 4). From October, the pronounced maximum over the
Norwegian Sea appears again with minimal contributions from land.

The gradual transition from more North Atlantic, North Sea, Norwegian Sea and Barents Sea contributing moisture sources
in winter to more local and continental Scandinavian and Eurasian contributions in summer can be partially explained by
changes in the geopotential height of the 500 hPa surface (Fig. 5). The zonal geostrophic flow south of Greenland is stronger
in winter than in summer, as shown by the stronger gradient of the geopotential height. The westerly zonal flow weakens in
summer, specifically in June, which could explain why June has the smallest and least variable precipitation.

Another way to describe moisture transport is to look at the integrated water vapour transport (IVT, mean annual cycle in
Fig. 6). Moist air masses from the North Atlantic are transported northeastward to the Scandinavian coast. By the influence
of polar easterlies, moist air masses over the Norwegian Sea seem to be transported in the direction of northeast Greenland.
This emphasises why the maximum of moisture source contribution is diagnosed over the Norwegian Sea for most months.
Evaporation over the Arctic Ocean seems to be prevented by sea ice and in summer, a gradual transition occurs towards more
IVT in the Arctic. In June, IVT is larger near to the study region, which is an indicator for the more local moisture sources
found by the Lagrangian moisture source diagnostic (Fig. 5). Furthermore, from July till September there is generally larger
IVT over the Eurasian continent. This coincides with the large fraction of contributing moisture sources over the north Eurasian
continent found by the Lagrangian diagnostic in these months.



**Figure 5.** Annual cycle of mean monthly attributed moisture sources contributing to precipitation in study region over the period February 1979–May 2017 (gridpoints coloured after their contribution). The mean sum of moisture sources over all gridpoints (i.e. accounted precipitation) is given for each month with its $95\,\%$ confidence interval and is only a part of the total precipitation (Fig. 4). The averaged mean $500\,\mathrm{hPa}$ geopotential height (grey lines) and the mean ice area cover (grey shaded area) are depicted as well.







**Figure 6.** Annual cycle of integrated water vapour transport (ERA-Interim monthly mean of February 1979–May 2017) [40° N–90° N].
The magnitude of IVT is depicted by the colours and the arrows indicate the IVT direction. The same scale is applied for each month. For
gridpoints with IVT$< 30\,\text{kg}\,\text{m}^{-1}\,\text{s}^{-1}$ only, the arrow length depicts also the IVT magnitude.



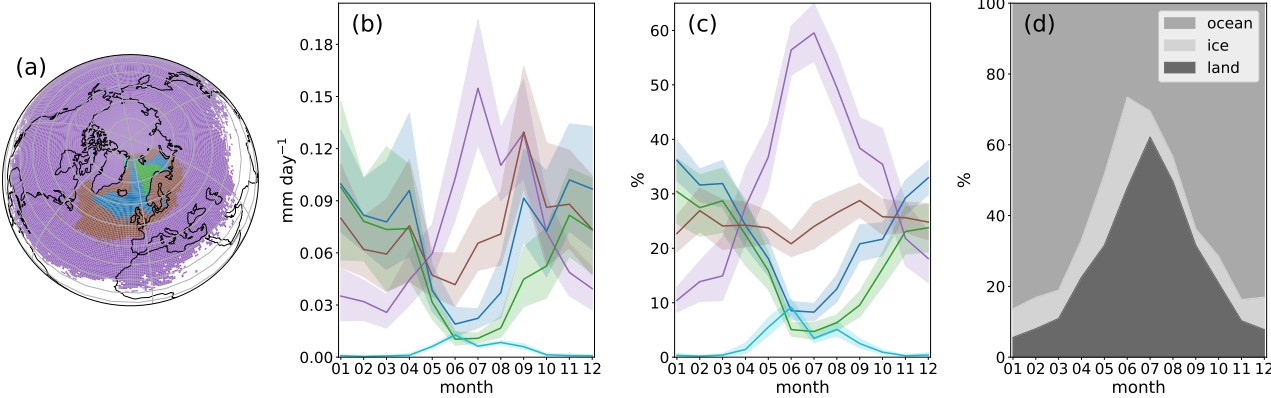

**Figure 7. (a)** K-means clustering of gridpoints into five clusters. The annual cycle of mean contributing moisture sources to the precipitation in the study region is plotted for each cluster in **(b)** absolute and **(c)** relative number. Shaded areas represent the 95 % confidence interval of the mean. In **(d)**, relative moisture source contributions from the ice-free ocean (sea ice concentration <0.5), the sea ice (sea ice concentration ≥0.5) and the land areas (same thresholds for land/sea mask) are depicted.

### 3.2.1 K-means clustering of moisture contributions

To analyse regional contributions of moisture sources, different moisture source regions were defined by applying a classi-
fication algorithm (K-means clustering, e.g., Wilks, 2011). K-means clustering separates data in samples grouped after their
similarities. In our case, we estimated similarity by first selecting the gridpoints that have contributed moisture sources over
the study period and then computing the percentage of each gridpoint's moisture source contribution to the total mean precipi-
tation for each month of the year. Therefore, a table of 24051 gridpoints x 12 months (where $\sum$gridpoints=100 %) was fed to
the algorithm (here: *sklearn*, Pedregosa et al., 2011). The algorithm then separates the gridpoints in a user-chosen number of
clusters, here based on the annual cycle of moisture source contribution to precipitation in the study region.

The raw output of the K-means clustering is plotted in Fig. 7a. Although the gridpoints' locations were not included in the
algorithm, the clusters mostly cover homogeneous areas, which means that gridpoints that are near to one another display
a similar behaviour in the relative moisture source contribution throughout the year. The algorithm recognises the features
of cluster formations from Fig. 5: the *green* coloured cluster corresponds to the area of a pronounced maximum in moisture
sources for most months, and the *cyan* coloured cluster corresponds to the local sources in summer directly above the study
region. The K-means clustering algorithm separated the gridpoints into clusters displaying significantly different behaviour
with respect to the 95 % confidence interval (Fig. 7b, c). The number of five distinct clusters shown here was chosen because
it produced the best compromise between differentiating behaviour patterns and still having significantly different clusters.
Another algorithm, spectral clustering (also available in *sklearn*, Pedregosa et al., 2011), produced similar results.

In winter, moisture sources over land contribute minimally (in January ∼6 %), however, in summer, the majority of moisture
sources come from land regions (in July ∼62 %). The moisture source contribution of sea ice areas is relatively low, but highest
in June (23 %, Fig. 7d). As June is the driest month with the highest contribution of local moisture sources (Fig. 7), there is an





indication that evaporation over sea ice near to the study region is contributing to precipitation in the study region. However, if those gridpoints are chosen that are defined with a sea ice concentration equal or above 0.9 (instead of 0.5, see Sect. 2.1), the contributions in all months decrease to a maximum of $15\%$ in June and is in most other months around $3\%$ (not plotted). Hence, large parts of contributing evaporation over a defined sea ice area occur over those gridpoints where the total area of the gridpoint is partially sea ice covered.

For more detailed analyses (and because the clustering algorithm cannot separate ocean from land sources), we now further refine the automated clusters with manual intervention. The *blue* coloured cluster in Fig. 7a does not differ between the Norwegian/Greenland and the Barents Sea moisture sources. To interpret the results in the context of NAO, we split the former *blue* coloured region into two regions (*2O* and *4O*) and distinguished between ocean (with sea ice) and land regions (compare Fig. 7a and Fig. 8a). Moreover, we divided the large former *violet* coloured cluster of Fig. 7a into two new groups: the *7O/7L* cluster that contributed least (in total only $10\%$ of the former *violet* coloured cluster area) and into the *6O/6L* cluster that contributed most (in total $90\%$ of the former *violet* coloured cluster area). This gives a better impression of areas contributing that are far away from the study region. Hence, the new separation results in seven ocean and four land clusters (Fig. 8a, the small *4L* cluster area is not a significant contributor and therefore neglected).

For the ocean clusters *2O, 3O, 4O* & *5O*, the relative maximum is in winter while the minimum is in summer (Fig. 8e). The Norwegian Sea (*3O*, Fig. 8) is one of the main moisture sources, specifically during winter. The Norwegian Sea is located below the North Atlantic storm track and is a main region for convective warming (Tsukernik et al., 2004) because of the relatively warm ice-free ocean and the relatively dry and cold air above resulting in the highest total column vapour (precipitable water) in the Arctic. In addition, in the colder seasons, more evaporation occurs because of larger vertical humidity gradients and stronger moisture transport due to higher temperature gradients between the subtropics and the Arctic (Serreze and Barry, 2014).

All land clusters have their maximum contribution in July except for the local *1L* cluster where the maximum occurs in June (Fig. 8d, g). A large part of summer Arctic precipitation comes from evapotranspiration over nearby land regions by regional recycling of water vapour that peaks in summer due to enhanced convection from stronger solar insolation (Serreze and Barry, 2014). The moist continental air masses from non-local regions over the north Eurasian continent are transported in summer towards northeast Greenland by a cyclone with a trough axis between Iceland and Svalbard (see 500 hPa geopotential height in Fig. 5 and IVT of Fig. 6). Large parts of these land clusters (*1L, 5L, 6L*, Fig. 8a) and the study region itself are located in the continuous permafrost zone (Brown et al., 1998). Thus, enhanced evapotranspiration in summer could also be explained by thawing of the uppermost permafrost layers (Biskaborn et al., 2019).

## 4 Changes to precipitation characteristics and moisture sources

### 4.1 Interannual variability from the North Atlantic Oscillation (NAO)

The NAO is one of the most important patterns of atmospheric circulation variability over the middle and high latitudes, specifically in the cold season (November–April; Hurrell et al., 2003). In its negative phase (NAO−), there is a weaker subpolar





**Figure 8. (a)** K-means clustering with additional manual separation. The annual cycle of moisture sources contributing to precipitation in the study region is plotted for each cluster in **(b, c, d)** as absolute number and in **(e, f, g)** as relative number in % of the total diagnosed moisture source amount. The ocean (with sea ice) regions in **(b, c, e, f)** are separated from the land regions in **(d, g)**, and land regions have in **(a)** more transparent colours. We use two graphs for the ocean regions for readability; grey lines in (b, c, e, f) correspond to the missing ocean regions for comparison. Shaded areas represent the 95 % confidence interval of the mean. Summing up the different regions of relative contribution for ocean (e, g) and for land (g) would give the total land or ocean (with & without sea ice) contribution shown in Fig. 7d. In **(h)**, the used cluster abbreviations and descriptions of approximate geographical regions of the K-means clusters of (a) are listed.

low over Iceland and a less pronounced subtropical high over the Azores, while in its positive phase (NAO+), a larger pressure gradient leads to stronger northeastward-oriented surface winds over the North Atlantic. In the following, we assess whether variability in the NAO affects the inter-annual variability of precipitation and moisture sources of the study region.



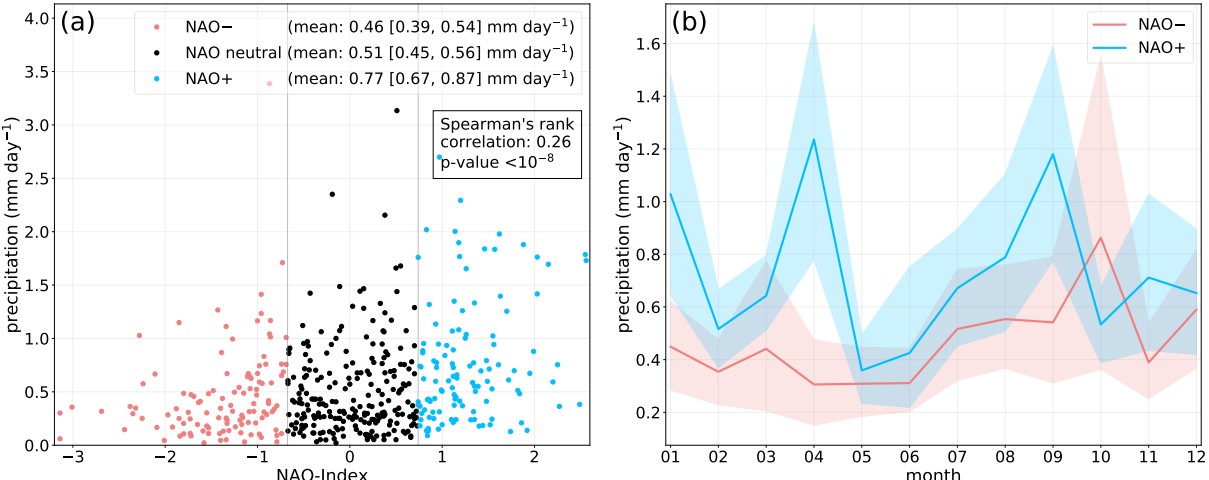

**Figure 9. (a)** Scatterplot of precipitation in study region against the NAO index for in total 460 months (February 1979–May 2017). Months were separated into months with NAO indices below or equal to the 25 % percentile (NAO−), above or equal to the 75 % percentile (NAO+), or in between the 25 % and 75 % percentile (NAO neutral). **(b)** Mean annual cycle of precipitation in study region for months with NAO being below or equal to the 25 % percentile (NAO−) and above or equal to the 75 % percentile (NAO+). For each month of the year, month-specific 25 % and 75 % percentile thresholds were computed. The shaded areas represent the 95 % confidence interval of the mean.

### 4.1.1 Relationship between NAO and precipitation

250 Generally, precipitation in the study region increases with increasing NAO index and is more variable (Fig. 9a). Mean precipitation with NAO+ is larger than with NAO− (Fig. 9a), specifically for January and April (Fig. 9b). As expected, variability in summer precipitation is not driven by NAO variability (when NAO is at its weakest). The month with the largest variability, September (Fig. 4), shows a non-significant increase in precipitation for months with NAO+ compared to those with NAO− (Fig. 9b).

### 4.1.2 Relationship between NAO and moisture sources

255 We start by analysing whether there are differences in the contributing moisture sources of the study region for NAO+ versus NAO− months for each individual cluster region separately over the annual cycle (Fig. 10). Significant differences between the NAO phases could only be found for some regions in January, April and September (Fig. 10), thus, these months were investigated in more detail (Fig. 11). For January NAO+ months, evaporation and moisture transport to the study region was 260 stronger from the eastern Norwegian Sea and Barents Sea (Fig. 11a), which corresponds mainly to the *3O* and *4O* cluster (Fig. 8a). In those clusters, significant differences between NAO− and NAO+ were found only for January (Fig. 10). For April and September NAO+ months, evaporation and moisture transport to the study region was enhanced from large parts of the North Atlantic above 45° N and of the ice-free Atlantic sector of the Arctic Ocean (Fig. 11b, c). Moisture source dependence on NAO in April was largest in the *2O*, *3O*, *4O*, and *5O* ocean clusters as well as the *5L* land cluster (Fig. 10). In September, the *2O*



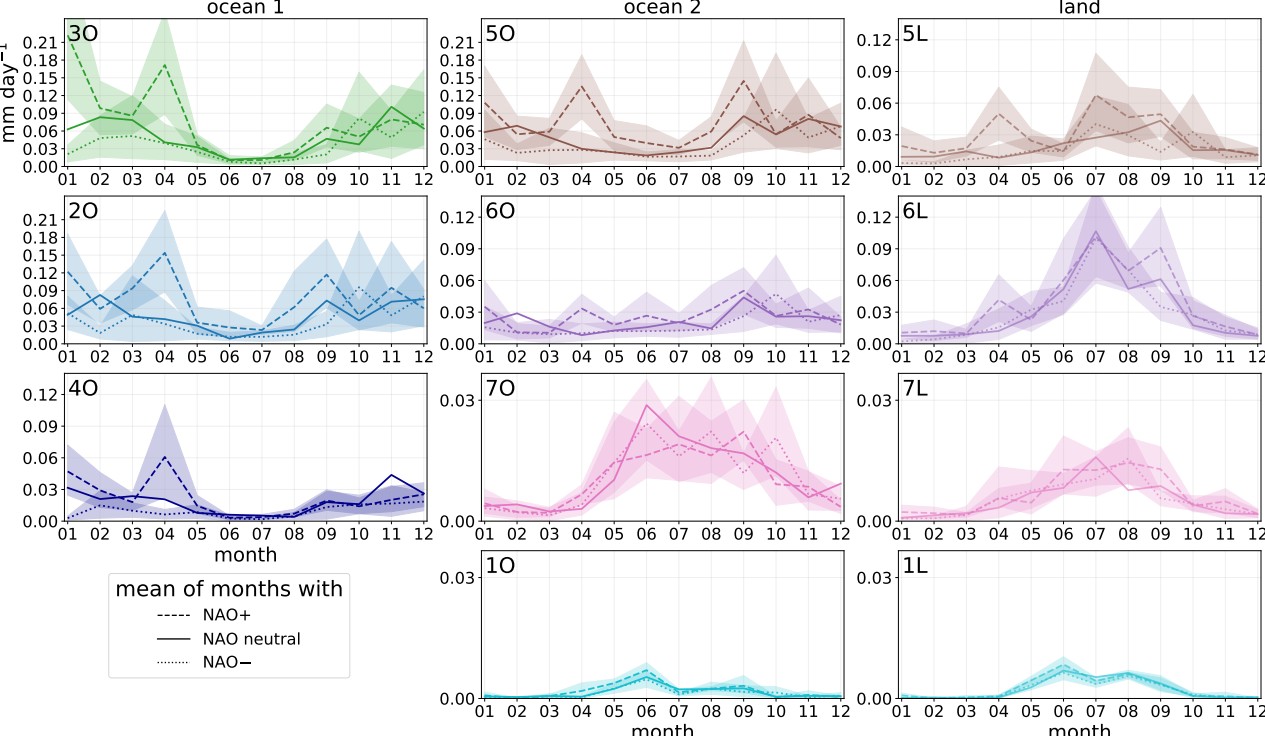

**Figure 10.** Mean annual cycle of contributing moisture sources distinguishing between months with NAO indices above or equal the 75 % percentile (NAO+), below or equal the 25 % percentile (NAO−) and those in between (NAO neutral) for the different clusters. The same thresholds as in Fig. 9b were chosen. Shaded areas represent the 95 % confidence interval of the mean of months for the highest and lowest NAO index quartile (February 1979–May 2017). The legend for the used cluster abbreviations is in Fig. 8a, h.

ocean as well as the *5L* and *6L* land moisture source clusters contributed significantly more for NAO+ than for NAO− months (Fig. 10). The larger NAO dependency of the *4O* cluster, part of Barents Sea, compared to the *2O* cluster, part of northeast Atlantic and western Norwegian Sea (Fig. 10), is another justification for the manual splitting of these areas that were clustered as one region by the K-means clustering (compare Fig. 7a and Fig. 8a). To conclude, there was an increased moisture uptake and transport to the study region for NAO+ months in January, April and September from the North Atlantic above $45°$ N and the ice-free Atlantic sector of the Arctic Ocean, specifically from the Norwegian Sea, which resulted in more precipitation over the study region for these months in the NAO+ phase.

## 4.2 Relationship to sea ice

A clear decreasing sea ice trend north of $30°$ N has been observed for the last 40 years. From ERA-Interim data, we compute $0.35 [0.26, 0.43]$ million $km^2$ $decade^{-1}$ yearly minimum sea ice area decrease (mostly September) and $0.67 [0.54, 0.80]$ million $km^2$ $decade^{-1}$ yearly maximum sea ice area decrease (mostly March). Bintanja and Selten (2014) showed that decreasing sea





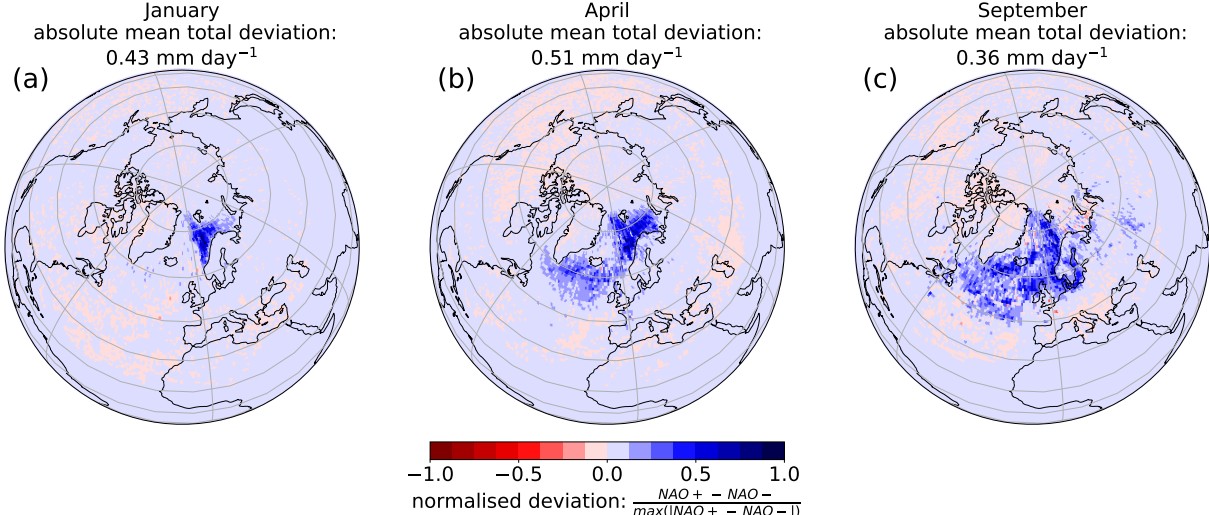

**Figure 11.** Normalised moisture source deviation between months with NAO indices being above or equal the $75\,\%$ percentile (NAO+) and months being below or equal the $25\,\%$ percentile (NAO−). The same thresholds as in Fig. 9b were chosen.

ice will enhance future evaporation in the Arctic, as open water at freezing point will replace ice at temperatures far below zero. We now test the working hypothesis that reduced sea ice results in larger contributing moisture sources for our study region.

For total precipitation in the study region, no significant correlation to the Arctic sea ice area was found when looking at each month of the year separately. Adding a time lag between sea ice and resulting precipitation of plus one or two months also resulted in no significant correlation. When comparing the seasonal mean precipitation against the maximum sea ice area of that year, we found some significant but small relations in autumn and winter.

A clearer insight might emerge when looking at the relation of each ocean cluster's attributed moisture sources against the respective relative sea ice area. When considering the total attributed moisture sources for each ocean cluster across the whole year, no significant correlations (Spearman's rank correlation with p-value<0.05) between moisture sources and relative sea ice area are found (Fig. 12a). If individual months are considered for each ocean region, then few significant correlations are observed. The main exception is for December, where the majority of regions display a significant correlation with relative sea ice area (Fig. 12a). Changes in the sea ice area can only poorly describe the variance in the moisture sources of entire ocean cluster areas (Fig. 12a) because large parts of them never had sea ice from 1980 till 2016 (for that month of the year or even not at all). Thus, the effect of sea ice in a given area on the attributed moisture source was further investigated by considering sub-regions of clusters with only those gridpoints that had once over the study period a sea ice concentration of above 0.5 (Fig. 12b). As this is different for each month of the year, for each month a different fraction of the ocean cluster was analysed. Compared to Fig. 12a, Fig. 12b shows that the annual contribution from sea ice related fractions of the *4O, 5O* and *6O* clusters is significantly correlated to decreasing sea ice. In addition, some more correlations for individual months were found over those specific fractions of the clusters (strongest in autumn-winter months, Fig. 12b). However, moisture sources over the sea



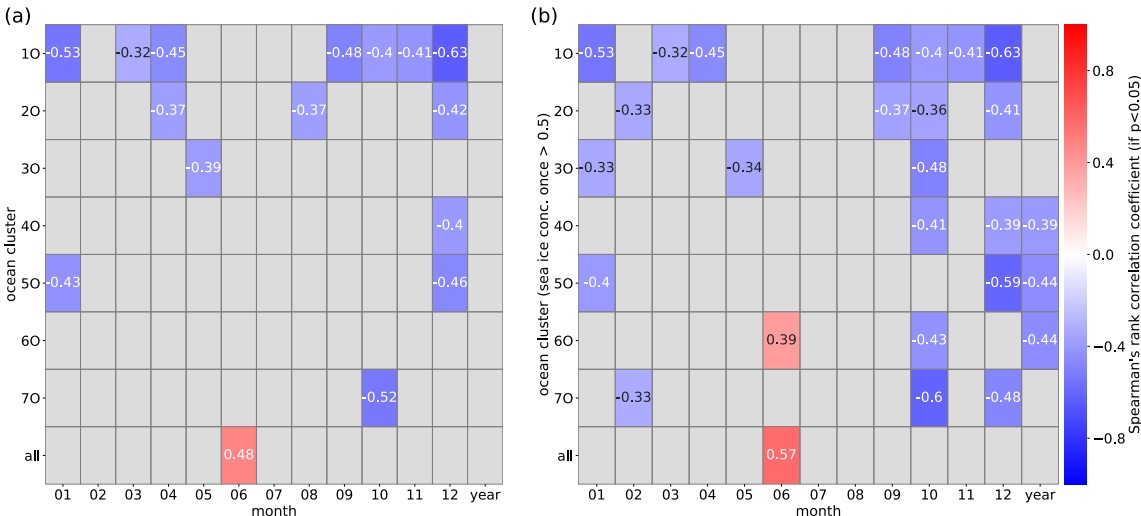

**Figure 12. (a)**: Spearman's rank correlation coefficients between moisture sources from each ocean cluster against relative ice area for each month and year. If there is no significant correlation (p-value≥0.05), the corresponding box is shaded in grey. **(b)**: same as in (a) but looking only at the area of those ocean gridpoints that had once a sea ice concentration of above 0.5 during the study period (effectively reducing each cluster's area to the sea ice relevant areas). The legend for the used cluster abbreviations is in Fig. 8a, h.

ice related sub-regions as defined in Fig. 12b contribute on average only 16 % to the entire diagnosed moisture sources. Hence, the correlations of moisture sources against sea ice (Fig. 12b) describe only a very small fraction of the entire moisture sources for the study region, which also explains why we did not find correlations between Arctic sea ice area and precipitation in the study region.

When specifically considering moisture source regions, the *1O* ocean cluster (closest to the study region) displays significant
correlations for seven months (September till April) with increasing attributed moisture sources over *1O* for decreasing relative sea ice area (Fig. 12a, b). Changes in the sea ice amount in *1O* change the general evaporation over the area, possibly directly influencing precipitation in the study region. For clusters located further away, changing sea ice might also directly effect the evaporation over that area. However, contributing moisture sources depend also on the moisture transport to the study region, which changes with decreasing sea ice as well. This might be one reason for the weak relations that we found. Looking at all
ocean clusters together, we only found a correlation for June, which was positive: this is not expected and is likely a statistical coincidence.

## 4.3  Temporal evolution

According to ERA-Interim, the study region has warmed by $2.8\,[1.6, 4.0]\,°\mathrm{C}$ (two metre temperature) in the 40-year period 1979–2018. We now test whether such a trend is also detectable for precipitation (or regional moisture sources) by looking
at its temporal evolution (Fig. 13a). A possible trend was tested by computing the Pearson correlation coefficient through a linear fit between time and precipitation or moisture sources. Linear regression analysis requires that residuals from the



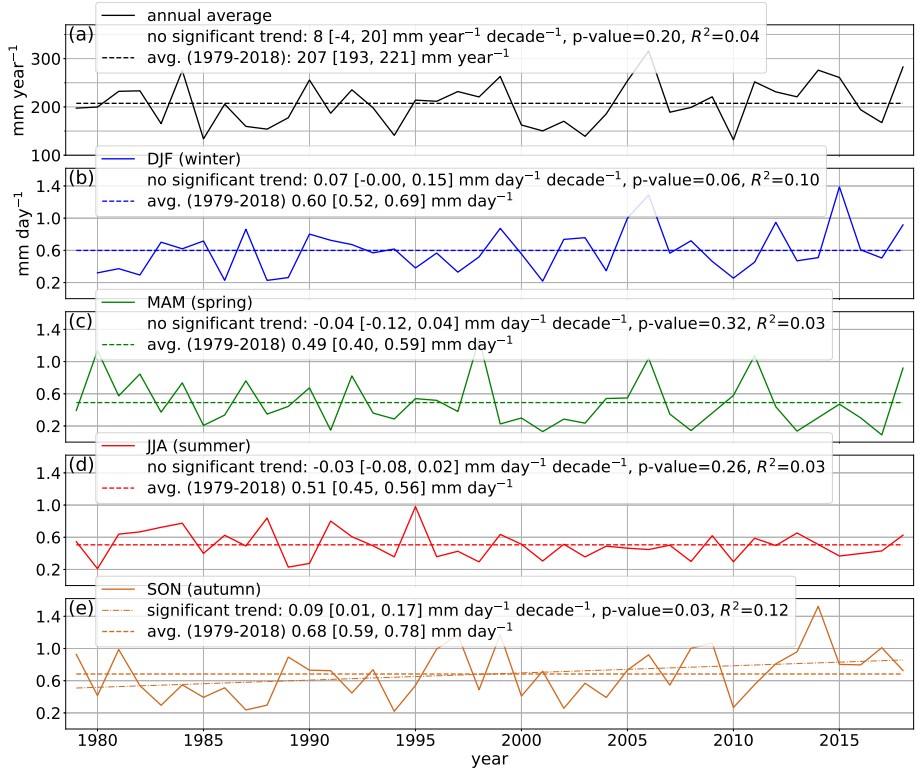

**Figure 13. (a)** Annual and **(b-e)** seasonal temporal evolution of precipitation amount in the study region from ERA-Interim with estimates of a possible linear trend. In autumn (SON), the existence of a positive upward linear trend can not be rejected under the $6\,\%$ level.

fitted regression line are normally distributed, which is not always the case for monthly data. Therefore, the more robust non-parametric Mann-Kendall trend test was also applied to detect whether a monotonic upward (downward) trend had occurred, which does not necessarily need to be linear (Wilks, 2011). The yearly, winter, spring and summer precipitation from 1979 315 to 2018 at the study region do not show a significant trend. There is a small increasing trend in autumn precipitation of $0.09\,[0.01, 0.17]\,\mathrm{mm\,day^{-1}\,decade^{-1}}$ (significant at the $5\,\%$ level for both the linear regression and Mann-Kendall trend tests).

Of specific interest is the changing contribution of different moisture source regions to the precipitation of the study region over time (1980–2016). Over the annual mean, no temporal trends were identified from any of the land or ocean clusters and also not from the relative land-ocean moisture source contribution. When looking at the monthly time series, some sporadic 320 slight trends are visible, however, they are too small in absolute numbers to be further considered. For example, in October, the only month with a small overall detectable increasing precipitation trend, very small significant trends of increasing contributing moisture sources with time were detected for the *7O* cluster and for the overall land contributions. This is also in-line with the increasing attributed moisture sources that were found for decreasing sea ice over the *7O* cluster in October (Fig. 12a, b).

The possible reasons for the absence of precipitation trends, despite the observed increase in temperature and loss of Arctic 325 sea ice during recent decades (Comiso and Hall, 2014) are discussed in Sect. 5.3.





## 5 Discussion

### 5.1 Moisture source regions and relation to the NAO

In Sodemann et al. (2008a), the majority of moisture sources (>85%) of the northern and east-central Greenland Ice Sheet are over the North Atlantic and Nordic Seas above $35°$ N, similar to our findings. In Nusbaumer et al. (2019), Greenland moisture

sources were estimated by water tracers using the Goddard Institute for Space Studies climate model and MERRA2 horizontal winds (mean of 1980–2015). They also found similar results: the dominant moisture source in northeast Greenland is the North Atlantic and the ice-free Atlantic sector of the Arctic Ocean except for summer (JJA) where continental sources are substantial.

For NAO+ winter months, Sodemann et al. (2008a) found that moisture sources for the northern and east-central Greenland Ice Sheet are larger over the Norwegian Sea, which is qualitatively similar to our findings. In case of NAO− winter months,

moisture sources were found to be further southward (maximum over $40°$ N–$60°$ N), which does not agree with our study, possibly because of the different regions considered. The same general relationship of increasing precipitation for higher NAO indices was found over the ice sheet by Sodemann et al. (2008a) and Koyama and Stroeve (2019). With a high NAO index, the pressure gradient between the subpolar low and subtropical high is larger, specifically in winter months, which results in stronger westerlies and increased intensity and number of storms in Iceland and the Norwegian Sea (Hurrell et al., 2003).

This shift in the North Atlantic storm activity explains the larger moisture sources over the Norwegian Sea for NAO+ and the positive correlation between NAO index and precipitation amount.

### 5.2 Relation to sea ice

Bintanja and Selten (2014) predicted a relative change of Arctic precipitation per degree surface temperature warming of $4.5\,\%\,K^{-1}$, which is larger than the global rate ($1.6$–$1.9\,\%\,K^{-1}$), due to feedback mechanisms associated with retreating winter

sea ice. For our study region, we did not yet observe significant amplified precipitation sensitivity for the recent decades, but our estimates of 3.8 [-1.6, 9.3] $\%\,K^{-1}$ over the annual mean do not contradict Bintanja and Selten (2014). We also found slight significant correlations of increasing precipitation for higher surface temperatures for January, February, March, September and November. A more in-depth analysis could look at temperature changes from the top of the inversion height or from the actual moisture source location. According to our analyses, only $16\,\%$ of the diagnosed moisture sources come from the

sea ice relevant sub-regions, which might explain why we found only weak correlations between precipitation and sea-ice extent. In Svalbard, for example, more moisture comes from regions where sea ice loss over the last decades was largest, e.g., Barents/Kara Sea (Faber et al., 2017), and the influence of changing sea-ice might be stronger there.

### 5.3 Temporal trend of precipitation and moisture sources

According to Bintanja and Selten (2014), precipitation will increase around $50\,\%$ (RCP 8.5 scenario, $25\,\%$ for RCP 4.5) in

northeast Greenland based on the differences between the means 2006–2015 and 2091–2100. The reasons are a strong increase in local surface evaporation through Arctic warming and retreating sea ice and to a lesser degree enhanced moisture inflow from





lower latitudes. We did not find a temporal trend in annual precipitation for the 40-year period 1979–2018. At Danmarkshavn (Fig. 1), no precipitation trend was found for the period 1981–2012 either but a significant trend (p-value<0.05) was found for 1971–2000 with $36\,\mathrm{mm\,year^{-1}\,decade^{-1}}$ and for 1961–1990 with $48\,\mathrm{mm\,year^{-1}\,decade^{-1}}$ (Mernild et al., 2015). From the

reanalysis products ASRv1 and ASRv2 for 2000–2012, no precipitation trend was visible for northeast Greenland and also not from observations for Danmarkshavn or Station Nord for 2000–2007 (Bromwich et al., 2016; Koyama and Stroeve, 2019).

Some of the inter-annual variability of precipitation occurs because of variability in the NAO. NAO has decreased in summer since the 1990s and winter NAO variability has increased (Hanna et al., 2015). This might be a reason why a significant annual temporal trend in the precipitation of the study region is not yet visible (Fig. 13a). According to Hurrell and Deser (2010),

since 2001, there have been more winter days with strong anticyclonic ridges over Scandinavia ("Blocking") and over western Europe ("Atlantic Ridge" regime) compared to NAO+ or NAO−, hence, differentiating between four regimes could improve the analysis.

We found a small trend of increasing precipitation over the 40-year period 1979–2018 for autumn (Fig. 13e). October is among the months where relations were observed between increasing moisture source contributions and decreasing sea ice in

some of the ocean clusters (Fig. 12b). It is also the month with the largest temperature increase of $1.5\,[0.9, 2.1]\,^{\circ}\mathrm{C\,decade^{-1}}$ compared to $0.7\,[0.4, 1.0]\,^{\circ}\mathrm{C\,decade^{-1}}$ in the annual mean for 1979–2018. This is consistent with future predictions of Bintanja and Selten (2014) where autumn is the most sensitive season with the largest predicted precipitation increase.

### 5.4 Limitations

Precipitation is difficult to quantify, specifically at higher latitudes and in remote areas (Serreze and Barry, 2014). No direct

precipitation measurements exist that could have been compared to the reanalysis data. The nearest observational stations are exposed to a more maritime climate (Station Nord & Danmarkshavn, Fig. 1), but the study region is inland over heterogeneous terrain at around 740 m a.s.l., where orographic uplift of moist air masses might alter local precipitation (Serreze and Barry, 2014). Precipitation measurements (corrected for undercatch) of Station Nord and Danmarkshavn show a good agreement with the Arctic System Reanalysis (ASRv1, Koyama and Stroeve, 2019). ASRv1 and ERA-Interim have similar precipitation

estimates in the study region (December 2006–November 2007, Bromwich et al., 2016), providing some confidence in the ERA-Interim estimates.

Besides the uncertainties from precipitation estimates, there are also limitations from the Lagrangian moisture source diagnostic. We showed in Sect. 2.2 that only $48.3\,\%$ of precipitation could be attributed to moisture sources. The remaining sources were detached from the surface (moisture uptake above the PBL) or were unidentifiable. This number is lower than in compara-

ble studies, and could be explained by the dry conditions in the region. Parametrised convection could also be responsible for a significant amount of vertical moisture transport and increases the non-accounted moisture uptake during summer (Sodemann and Zubler, 2010). Not distinguishing between below and above PBL height moisture uptake could be a way to increase the attribution (Fremme and Sodemann, 2019), at the cost of larger uncertainties since moisture uptake above the PBL cannot be assigned to a specific location. A direct comparison to other models would be necessary to estimate uncertainties resulting

from the choice of the Lagrangian model itself (Van Der Ent and Tuinenburg, 2017; Winschall et al., 2014).





## 6 Conclusions

We analysed the present-day moisture sources for a region in northeast Greenland at $80°$ N, a polar desert with a mean annual precipitation of $207\,[192, 224]$ mm year$^{-1}$ (1979–2018, ERA-Interim).

The main moisture source region is the North Atlantic above $45°$ N and the ice-free Atlantic sector of the Arctic Ocean
with a maximum over the Norwegian Sea, which is largest for months in the NAO+ phase, specifically in January and April. This leads to stronger and more variable precipitation in the study region for these months. While the main moisture sources are over the ocean in winter months, in summer the contributions from land regions (locally or north Eurasian continent) are largest (60 % in July). The month with the highest precipitation is September (contributions from both land and ocean moisture sources), whereas the month with the least precipitation is June (mostly land sources).

The study region has warmed by $2.8\,[1.6, 4.0]$ °C and surrounding Arctic sea ice has retreated for the 40-year period considered. The amount of moisture uptake (and transport) from sea ice related regions increased with decreasing sea ice for the study region, specifically in October and December. Thus, one might expect to see already an increasing trend in precipitation in the study period. However, as most moisture source contributions come from permanently ice-free ocean regions and because of the large inter-annual variability from the NAO, we could not detect considerable trends in precipitation in the study region. To
better understand the underlying mechanisms, future studies could focus on the pathway of moisture source transport during extreme precipitation events, which account for a large part of total precipitation.

Longer time periods need to be considered for more robust results. The acquisition and analysis of palaeoclimate proxies might yield further insights into the long-term climate dynamics of the region, thus further providing a baseline and enabling improved predictions in this highly sensitive region in the future.

*Data availability.* The ERA-Interim reanalysis data used in this study can be accessed from the European Center for Medium-Range Weather Forecasts (ECMWF; https://www.ecmwf.int/en/forecasts/datasets/reanalysis-datasets/era-interim Dee et al., 2011). The monthly moisture source estimates from the Lagrangian diagnostic (February 1979 – May 2017) are made publicly available via Zenodo: https://doi.org/10.5281/zenodo.3972882.

*Author contributions.* LS undertook the majority of the analyses, interpreted the data, and wrote the majority of the manuscript. LL undertook
the computation of the Lagrangian moisture source diagnostic. GM and FM conceived the project and wrote parts of the manuscript. All authors directly contributed to the manuscript through discussion or writing.

*Competing interests.* The authors declare that they have no competing interests.



*Acknowledgements.* We would like to thank Harald Sodemann for helpful discussions during the planning phase. This research has been supported by the Austrian Science Fund (grant no. Y 1162-N3) and with start-up funding from the University of Innsbruck Research Centre

for Climate – Cryosphere and Atmosphere, both to Gina E. Moseley.





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
