# Peer review of "Lagrangian detection of precipitation moisture sources for an arid region in northeast Greenland: relations to the North Atlantic Oscillation, sea ice cover and temporal trends from 1979 to 2017"

_Weather and Climate Dynamics, 2020_

## Referee Comment (RC1) · Anonymous Referee #1 · 23 Oct 2020

Summary

This manuscript uses the ERAinterim reanalysis in combination with a Lagrangian di-
agnostic to investigate precipitation and moisture sources for a small arid region in
northeast Greenland during the years 1979 to 2017. The results show a strong sea-
sonal cycle in moisture sources, with dominant contributions from the North Atlantic
and Arctic Ocean in winter, and from local sources and Eurasia in summer. In contrast
to the temperature and sea ice trends, the authors found no significant temporal trends
in precipitation or moisture sources, apart from a slight positive trend for precipitation

in autumn. They showed that the North Atlantic Oscillation (NAO) can explain some of the variability: NAO+ leads to more and more variable precipitation in the study region and more moisture transport from the Norwegian Sea than NAO-.

The manuscript helps to place paleoclimate records from northeast Greenland into the context of present-day climate (change). It is well written and has a clear structure, and the figures are very nice and easy to understand. I only have two general comments (see below), and recommend that the paper be published after minor revisions.

General comments

I assume that the diagnosed moisture sources would look different if different thresholds and/or time steps were chosen. For example, a shorter time step would probably lead to more local moisture sources, because more moisture losses would discount earlier moisture uptakes. The minimum moisture increase that counts as a moisture uptake (what is it?) might be important as well. It would be good to include some sensitivity tests (e.g. in the supplement) that quantify this, and how it affects the conclusions of the manuscript.

The low percentage of accounted precipitation (less than 50%) makes all the conclusions regarding moisture sources relatively weak. If possible, it would be good to increase this percentage somehow. If a large part of the moisture uptakes are unidentifiable because they occurred before the start of the trajectories, this could easily be achieved by running longer backward trajectories. Another idea (in line with the first general comment above) is to use shorter time steps by including the forecast data of ERAinterim, or by using the hourly ERA5 output instead of ERAinterim. This would likely shift the moisture sources closer to the study site and increase the percentage.

Specific comments

L21: Add references for this first sentence?

L92: Why three different time periods? This is a bit confusing (but a detail).

L155: fewer –> less

L157: less –> few

L167: Maybe write explicitly that this is not shown.

Figure 5: It looks like the geopotential height lines stop at 5700 (?)

Figure 6: Is the different map projection here on purpose?

L200: Are clusters calculated based on the absolute or relative moisture source contribution?

L210&211: Add Fig 7d in brackets.

L217: I wonder what the k-means algorithm would do for 11 clusters. Would they look similar to the manual clusters?

L222: There are no land regions for the former blue cluster → mention land regions later.

L233: Maybe mention also 6O and 7O.

Figure 8, caption: (e, g) –> (e, f)

L247: northeastward-oriented –> southwesterly

L252: What is meant by NAO is at its weakest? NAO-, or neutral?

Figure 10: Switch 3O and 2O?

Figure 11: Is the sum of all values zero (it does not look like)? If not, I am not sure how they were normalized.

Section 4.2: Suggestion: What I would find useful here is a figure showing the correlations on a map instead of in a table for the clusters.

L282+: What about evaporation alone? Did it increase with decreasing sea ice?

Figure 13: Is the p-value for the linear regression or Mann-Kendall test? Please clarify.

L347: This is a bit confusing, before only October was mentioned, but it was a different unit.
* * *

---

## Author Comment (AC2) · 25 Nov 2020

Please find our answers to reviewer #2 in the Supplement.

Please also note the supplement to this comment:
https://wcd.copernicus.org/preprints/wcd-2020-42/wcd-2020-42-AC2-supplement.pdf

---

## Author Comment (AC1)

We would like to thank the anonymous referee for taking the time to read our manuscript and provide constructive comments which helped us to improve our manuscript. We hope that our response is clarifying and we remain available for further questions.

Here we present a detailed point by point response (the reviewer's comments are given in italics, our answer in normal font). When appropriate, we indicate the text that has been added to the manuscript as a separate paragraph in quotation marks.

**General comments:**

**RC (summary):** This manuscript uses the ERA-interim reanalysis in combination with a Lagrangian diagnostic to investigate precipitation and moisture sources for a small arid region in northeast Greenland during the years 1979 to 2017. The results show a strong seasonal cycle in moisture sources, with dominant contributions from the North Atlantic and Arctic Ocean in winter, and from local sources and Eurasia in summer. In contrast to the temperature and sea ice trends, the authors found no significant temporal trends in precipitation or moisture sources, apart from a slight positive trend for precipitation in autumn. They showed that the North Atlantic Oscillation (NAO) can explain some of the variability: NAO+ leads to more and more variable precipitation in the study region and more moisture transport from the Norwegian Sea than NAO-. The manuscript helps to place paleoclimate records from northeast Greenland into the context of present-day climate (change). It is well written and has a clear structure, and the figures are very nice and easy to understand. I only have two general comments(see below), and recommend that the paper be published after minor revisions

**AR:** Thank you for the positive assessment of our study.

**RC:** I assume that the diagnosed moisture sources would look different if different thresholds and/or time steps were chosen. For example, a shorter time step would probably lead to more local moisture sources, because more moisture losses would discount earlier moisture uptakes. The minimum moisture increase that counts as a moisture uptake (what is it?) might be important as well. It would be good to include some sensitivity tests (e.g. in the supplement) that quantify this, and how it affects the conclusions of the manuscript.

**AR:** We agree that both the input data we are using (ERA-Interim) as well as the methodology (adapted from Sodemann et al., 2008a) are subject to uncertainties. ERA-Interim data is available at  $\Delta t$ =6h in the analysis fields and at the different model levels (shorter time steps of  $\Delta t$ =3h are only used for forecast surface parameters such as precipitation). Shortening the timestep with ERA-Interim would therefore imply interpolating between the 6H timesteps, which would introduce other uncertainties and with unclear added value. Similar studies (e.g., Langhamer et al., 2018; Fremme and Sodemann, 2019) also used at 6H timestep, and the method of Sodemann et al. (2008a) was developped and tested on a 6H step. However, we fully agree that a smaller time step and grid resolution could give a finer picture of moisture uptake, because the applied moisture source diagnostic needs the assumption of either evaporation or precipitation dominating in one time step. It should now be possible to realise such sensitivity studies using the newer ERA5 dataset (not yet available when we started our study) albeit with considerably increased data management and computational requirements. Such analyses would be very demanding at this stage, and we argue that this should be left for follow-up studies.

To emphasize the uncertainties mentioned by the reviewer, we added the following sentence into the "Limitations" section of the manuscript (Sect. 5.4):

"In the moisture source diagnostic, either evaporation or precipitation can occur in each time step of 6 h. Therefore, using shorter time steps and a finer grid resolution (e.g. using the ERA5 reanalysis dataset instead of ERA-Interim) could influence the diagnostic."

In our study we did not use any threshold for the minimum moisture increase, hence we set the threshold  $\Delta q_c^0$  that was used in Sodemann et al. (2008a) ( $0.2 g kg^{-1}$  with ERA-40 for Greenland winter moisture sources) to zero. According to Sodemann et al. (2008a), this threshold was necessary in their study to suppress spurious uptakes from numerical noise and reduced the computational cost. However, as we used ERA-Interim and had enough computational power, we decided to not use any threshold at all. We clarified this by adding the following into Sect. 2.2:

..."and we also did not use any minimum moisture uptake threshold in contrast to Sodemann et al. (2008a)."

**RC:** The low percentage of accounted precipitation (less than 50%) makes all the conclusions regarding moisture sources relatively weak. If possible, it would be good to increase this percentage somehow. If a large part of the moisture uptakes are unidentifiable because they occurred before the start of the trajectories, this could easily be achieved by running longer backward trajectories. Another idea (in line with the first general comment above) is to use shorter time steps by including the forecast data of ERA-interim, or by using the hourly ERA5 output instead of ERA-interim. This would likely shift the moisture sources closer to the study site and increase the percentage.

**AR:** Yes, you are totally right that accounting for less than 50% of precipitation is not the detection efficiency that we would have hoped for. The suggested extension of the backward trajectories has only a minor effect on the detection efficiency, e.g. in the case of Dec 1999 65% of moisture sources could be detected (below scaled PBL), 35% were above the PBL and only 5% is the preexisting moisture at the end of the backward trajectory. In addition, the trajectory density after 15 days backward is very low and spread out over a large surface. In the case of the South Patagonian Icefield, extending the trajectories backward to 20 days only diminished the preexisting moisture to 3%. The main reason for the weak detection efficiency is that we distinguish between moisture uptake below and above the scaled PBL height. Above the PBL one can not assume anymore that the air is well-mixed, so the moisture uptake there can not be directly assumed to be a result of evaporation occurring at the surface. There are studies, which

apply the same methodology and consider moisture uptake in the free atmosphere as well (e.g., Baker et al., 2015; Fremme and Sodemann, 2019; Hu et al., 2020), however it is unclear how they justify this approach. Another study uses the same approach but does not mention the detection efficiency (Bohlinger et al., 2017).

As discussed above, shorter time steps could also improve the percentage, however, this would need to be analysed in a a further study. A much better detection efficiency below the scaled PBL height was found when using lagrangian moisture source diagnostic of (Sodemann et al., 2008a) that we used but with dynamically downscaled ERA-20C reanalysis data (coupled COSMO-CLM+NEMO, personal communication with Amelie Krug, https://doi.org/10.5194/egusphere-egu2020-2315)

We argue however that we discussed this uncertainty at length in the manuscript (by describing it, discussing probable causes, and by comparing our values with previous studies). Furthermore, the fact that we observe a high correlation between precipitation amounts and accounted precipitation is an indicator that our variability analyses are robust.

**Specific comments:**

**RC: L21 Add references for this first sentence?**

**AR:** Thanks. We added "(Screen and Simmonds, 2010; Bintanja and Van der Linden, 2013; Bintanja and Selten, 2014; Bintanja and Andry, 2017)" into that sentence (see Sect. 1).

**RC: L92** Why three different time periods? This is a bit confusing (but a detail).**

**AR:** Yes, this is a bit confusing. However, we wanted to use the most out of each dataset we had. Moisture sources could not be computed for the full month of January 1979 because of the computation of the backward trajectories that would have needed data from December 1978. Therefore, we only computed moisture sources from February 1979 onwards. For moisture source trends we had to use only full years and therefore we had to shorten the time period.

**RC: L155 fewer $\rightarrow$ less**

**AR:** Thanks, we changed this as suggested.

**RC: L157** less  $\rightarrow$  few

**AR:** Thanks, we changed this as suggested.

RC: L167 Maybe write explicitly that this is not shown.

**AR:** Thanks for pointing that out. Because of specific comments of reviewer 2, we deleted this part of the paragraph.

**RC: Figure 5** It looks like the geopotential height lines stop at 5700 (?)**

**AR:** For July or August there was actually also a 5800 geopotential height line. As suggested by reviewer 2, we increased the amount of contour lines (every 50 m, but labelling only every 100 m). We hope that this makes it easier to distinguish the differences from month to month.

**RC: Figure 6:** Is the different map projection here on purpose?**

**AR:** Thanks. As reviewer 2 correctly noted the large white spaces by the orthographic projection, we switched to the North polar stereographic projection (on Fig. 5 and Fig. 11). For the legends of Fig. 7 and Fig. 8, we preferred the orthographic projections to show the full extent of the clusters.

**RC: L200** Are clusters calculated based on the absolute or relative moisture source contribution?

**AR:** The K-means clustering is based on the relative moisture source contribution. Maybe this was not clear enough, therefore we added the word relative into the description (see Sect. 3.2.1):

..., here based on the annual cycle of "relative" moisture source contributions to precipitation ...

**RC: L210/211 Add Fig 7d in brackets.**

**AR:** Thanks, we changed it as suggested.

**RC: L217** I wonder what the k-means algorithm would do for 11 clusters. Would they look similar to the manual clusters?

**AR:** What we aimed to do with the K-Means clustering is to find clusters/regions that have a similar behaviour over the annual cycle (e.g. all gridpoints of the brown region have a maximum in September and minimum in June). We also tried higher number of clusters, however in this case the differences between the seasonal cycles were not large enough and they did not give us further information (only very similar new clusters). The manual separation into land/ocean was necessary to better interpret the results. Another more complex approach could have been to first separate land and ocean, and then do some kind of K-means clustering where annual, NAO and sea ice variability are included.

**RC: L222** There are no land regions for the former blue cluster  $\rightarrow$  mention land regions later

**AR:** Thanks for the suggestion. We have restructured the paragraph and mention the distinction between land and ocean areas first. In addition, we clarified that the former blue cluster is an ocean region (both in Sect. 3.2.1).

**RC: L233** Maybe mention also 60 and 70

**AR:** Thanks. We added the following into Sect. 3.2.1:

"The ocean clusters 10, 70 peak in June and the 60 cluster peaks in October possibly as a consequence of more sea ice free areas in October."

**RC: Figure 8**, caption: (e, g)  $\rightarrow$  (e, f)

**AR:** Thanks, this was a typo. We changed it as you suggested.

**RC: L247** northeastward-oriented  $\rightarrow$  southwesterly

**AR:** Thanks. We changed it as suggested.

**RC: L252** What is meant by NAO is at its weakest? NAO-, or neutral?

**AR:** We meant with that NAO– and clarified this by writing instead (see Sect. 4.1.1):

"(when NAO is weakest, hence most negative)"

RC: Figure 10 Switch 30 and 20?

**AR:** Thanks, we switched the order of 30 and 20 as you suggested.

**RC: Figure 11** Is the sum of all values zero (it does not look like)? If not, I am not sure how they were normalised.

**AR:** To analyse the moisture source deviations between NAO+ and NAO- months, we subtracted for each gridpoint the moisture sources of the months with NAO- from NAO+. In order to better compare this between the months we divided each grid point by the maximum difference between the months with NAO+ and the months with NAO-. This means the gridpoint where there is the largest positive difference has a normalised deviation of 1. So, summing up all gridpoints times multiplying them with the maximum moisture source difference between NAO+ and NAO- gives the number that is written below the month as absolute mean total deviation of contributing moisture sources. We added this information in a shortened version to the legend and caption of Fig. 11.

**RC: Section 4.2** Suggestion: What I would find useful here is a figure showing the correlations on a map instead of in a table for the clusters.

**AR:** We have considered to present the results on a geographical map comparable to Fig. 8 with the respective correlation coefficients. However, this would result in 13\*2 subplots which would not improve the layout and readability of the results. Therefore, we decided to leave the table.

**RC: L282+** What about evaporation alone? Did it increase with decreasing sea ice?

**AR:** We expect that evaporation alone increases with decreasing sea ice indeed (e.g. described in Bintanja and Selten, 2014).

**RC: Figure 13** Is the p-value for the linear regression or Mann-Kendall test? Please clarify.

**AR:** Thanks for pointing this out. The p-values that are written in Fig. 13 are from the linear regression. We added the following into the caption of Fig. 13 to clarify this:

..." with estimates of a possible linear trend and its corresponding p-values".

**RC: L347** This is a bit confusing, before only October was mentioned, but it was a different unit.

**AR:** In this paragraph, we describe the months with a slight significant correlation of increasing precipitation for higher surface temperature. It is true that October is the only month where we could see a temporal trend (see Sect. 4.3)but when looking at the relation between precipitation and surface temperature there are other months with a significant correlation (see Sect. 5.2). We added into the subsection name (Sect. 5.2) where this paragraph is located ...Relation to "temperature" and sea ice... which might clarify that the first of the two paragraphs is about temperature.

**References**

- Baker, A. J., Sodemann, H., Baldini, J. U., Breitenbach, S. F., Johnson, K. R., van Hunen, J., and Pingzhong, Z.: Seasonality of westerly moisture transport in the East Asian summer monsoon and its implications for interpreting precipitation  $\delta$ 18O, Journal of Geophysical Research, 12O, 5850–5862, https://doi.org/10. 1002/2014JD022919, URL https://onlinelibrary.wiley.com/doi/10.1002/joc. 6781http://doi.wiley.com/10.1002/2014JD022919, 2015.
- Bintanja, R. and Andry, O.: Towards a rain-dominated Arctic, Nature Climate Change, 7, 263, https://doi.org/10.1038/nclimate3240, 2017.
- Bintanja, R. and Selten, F.: Future increases in Arctic precipitation linked to local evaporation and sea-ice retreat, Nature, 509, 479, https://doi.org/10.1038/nature13259, 2014.
- Bintanja, R. and Van der Linden, E.: The changing seasonal climate in the Arctic, Scientific reports, 3, 1556, https://doi.org/10.1038/srep01556, 2013.
- Bohlinger, P., Sorteberg, A., and Sodemann, H.: Synoptic conditions and moisture sources actuating extreme precipitation in Nepal, Journal of Geophysical Research: Atmospheres, 122, 12,653–12,671, https://doi.org/10.1002/2017JD027543, URL https://onlinelibrary.wiley.com/doi/abs/10.1002/2017JD027543, 2017.
- Fremme, A. and Sodemann, H.: The role of land and ocean evaporation on the variability of precipitation in the Yangtze River valley, Hydrology and Earth System Sciences, 23, 2525–2540, https://doi.org/10.5194/hess-23-2525-2019, 2019.
- Hu, Q., Jiang, D., Lang, X., and Yao, S.: Moisture sources of summer precipitation over eastern China during 1979–2009: A Lagrangian transient simulation, International Journal of Climatology, p. joc.6781, https://doi.org/10.1002/joc.6781, URL https:// onlinelibrary.wiley.com/doi/10.1002/joc.6781, 2020.

- Langhamer, L., Sauter, T., and Mayr, G. J.: Lagrangian Detection of Moisture Sources for the Southern Patagonia Icefield (1979-2017), Frontiers in Earth Science, 6, 219, https://doi.org/10.3389/feart.2018.00219, 2018.
- Screen, J. A. and Simmonds, I.: The central role of diminishing sea ice in recent Arctic temperature amplification, Nature, 464, 1334, https://doi.org/10.1038/nature09051, 2010.
- Sodemann, H., Schwierz, C., and Wernli, H.: Interannual variability of Greenland winter precipitation sources: Lagrangian moisture diagnostic and North Atlantic Oscillation influence, Journal of Geophysical Research: Atmospheres, 113, URL https://doi.org/10.1029/2007JD008503, 2008a.